# An MRI-Based Radiomic Prognostic Index Predicts Poor Outcome and Specific Genetic Alterations in Endometrial Cancer

**DOI:** 10.3390/jcm10030538

**Published:** 2021-02-02

**Authors:** Havjin Jacob, Julie A. Dybvik, Sigmund Ytre-Hauge, Kristine E. Fasmer, Erling A. Hoivik, Jone Trovik, Camilla Krakstad, Ingfrid S. Haldorsen

**Affiliations:** 1Section of Radiology, Department of Clinical Medicine, University of Bergen, 5020 Bergen, Norway; Julie.dybvik@uib.no (J.A.D.); sigmund78@gmail.com (S.Y.-H.); kristine.eldevik.fasmer@helse-bergen.no (K.E.F.); 2Mohn Medical Imaging and Visualization Centre (MMIV), Department of Radiology, Haukeland University Hospital, 5021 Bergen, Norway; 3Centre for Cancer Biomarkers, Department of Clinical Science, University of Bergen, 5021 Bergen, Norway; Erling.Hoivik@uib.no (E.A.H.); jone.trovik@helse-bergen.no (J.T.); camilla.krakstad@uib.no (C.K.); 4Department of Obstetrics and Gynaecology, Haukeland University Hospital, 5021 Bergen, Norway

**Keywords:** radiomics, endometrial cancer, MRI, molecular markers, LASSO regression, prognostic model

## Abstract

Integrative tumor characterization linking radiomic profiles to corresponding gene expression profiles has the potential to identify specific genetic alterations based on non-invasive radiomic profiling in cancer. The aim of this study was to develop and validate a radiomic prognostic index (RPI) based on preoperative magnetic resonance imaging (MRI) and assess possible associations between the RPI and gene expression profiles in endometrial cancer patients. Tumor texture features were extracted from preoperative 2D MRI in 177 endometrial cancer patients. The RPI was developed using least absolute shrinkage and selection operator (LASSO) Cox regression in a study cohort (n = 95) and validated in an MRI validation cohort (n = 82). Transcriptional alterations associated with the RPI were investigated in the study cohort. Potential prognostic markers were further explored for validation in an mRNA validation cohort (n = 161). The RPI included four tumor texture features, and a high RPI was significantly associated with poor disease-specific survival in both the study cohort (*p* < 0.001) and the MRI validation cohort (*p* = 0.030). The association between RPI and gene expression profiles revealed 46 significantly differentially expressed genes in patients with a high RPI versus a low RPI (*p* < 0.001). The most differentially expressed genes, *COMP* and *DMBT1*, were significantly associated with disease-specific survival in both the study cohort and the mRNA validation cohort. In conclusion, a high RPI score predicts poor outcome and is associated with specific gene expression profiles in endometrial cancer patients. The promising link between radiomic tumor profiles and molecular alterations may aid in developing refined prognostication and targeted treatment strategies in endometrial cancer.

## 1. Introduction

Endometrial cancer is the most common gynecologic malignancy in developed countries. The incidence rate of endometrial cancer has risen steadily during recent years, attributed to increasing obesity and high age in the population [1,2]. Surgicopathological staging according to the International Federation of Gynecology and Obstetrics (FIGO) staging system and histological subtyping is important for prognostication and guide choice of treatment [3]. Several studies have provided comprehensive molecular characterizations of endometrial cancers [4,5], including The Cancer Genome Atlas (TCGA) project identifying four prognostic groups in endometrial cancer based on gene expression profiles, suggesting a molecular-based reclassification of these tumors [6].

Pelvic magnetic resonance imaging (MRI) is the preferred imaging modality for local preoperative staging and routinely guides the choice of surgical procedure for endometrial cancer [7]. Radiomics involves high-throughput extraction of numerous quantitative imaging features using data-characterization algorithms [7,8,9,10]. Radiomic tumor features based on MRI have been shown to predict histological subtypes, treatment response, and outcome in various cancers including breast, rectal, cervical, and endometrial cancer [8,9,10,11,12]. Tumor texture analysis based on a manually defined tumor region of interest (ROI) on diagnostic images is a branch of radiomics where, e.g., voxel intensity histogram shapes are assessed [13]. Some textural radiomic features putatively reflect tumor heterogeneity and have been linked to an aggressive clinical phenotype in breast, gastric, and endometrial cancer [13,14,15,16].

Associations between radiomic features and the corresponding genomic tumor profiles have been reported in several cancers including breast cancer, glioblastoma, and non-small cell lung cancer [17,18,19]. In breast cancer, a study addressed the intratumor heterogeneity challenges by defining specific subclones and their biological functions based on radiogenomic signatures [20]. In glioblastoma, an MRI radiomics survival risk-score predicts the regulation of specific molecular signaling pathways responsible for chemotherapy response [21]. Importantly, cross-scale association analyses linking radiomic- and genomic tumor features may allow non-invasive identification of patients who are candidates for targeted treatment strategies based on their radiogenomic profile.

In the current study, we aimed to develop and validate a radiomic prognostic index (RPI) derived from textural features extracted from preoperative MRI and to explore possible associations between the RPI and the corresponding transcriptional profile in endometrial cancer patients.

## 2. Experimental Section

### 2.1. Patient Samples and Patient Cohorts

This study was conducted with written informed consent from all patients and under institutional review board approved protocols (REK Vest #2014/1907; 2015/2333) [22]. A population-based patient series diagnosed with endometrial cancer was prospectively collected from 2001 to 2014 in Hordaland Country (Norway). All patients underwent surgicopathological staging according to the international FIGO 2009 staging criteria [23], and the median follow-up time was 5 (0–12) years. Tumor samples were obtained from hysterectomy specimens and collected in the Bergen Gynecologic Cancer Biobank. Clinical characteristics and follow-up data were registered from the patients’ records as previously described [22].

A total of 338 patients were included in the study. The cohorts were designed based on availability of data. The study cohort comprised 95 endometrial cancer patients who had preoperative MRI (with subsequent manual tumor segmentation allowing tumor texture analysis) and tumor tissue for gene expression analysis (Figure 1, Table 1). The MRI validation cohort comprised 82 endometrial cancer patients who had MRI data with tumor texture analysis, whereas the mRNA validation cohort comprised 161 endometrial cancer patients who had gene expression data (Figure 1 and Table 1). There were no significant differences in patient characteristics between the three cohorts, except for surgicopathologically defined deep myometrial invasion being less frequent in the MRI validation cohort (in 34%) compared with the study cohort (in 50%) and mRNA validation cohort (in 51%; *p* = 0.04). In the study cohort, the mean (range) interval between MRI examination and tissue sampling for mRNA analysis at surgical staging was 11 (1–98) days.

The radiomic prognostic index (RPI) was developed in the study cohort (n = 95) comprising patients with available preoperative MRI data and gene expression data. The prognostic value of the RPI was validated in the MRI validation cohort (n = 82). Associations between gene expression and the RPI were analyzed in the study cohort (n = 95), and the prognostic value of the identified genes was validated in the mRNA validation cohort (n = 161).

### 2.2. MR Imaging and Texture Analysis

Preoperative pelvic MRI was performed on a 1.5T Siemens Avanto running Syngo MR B17 (Siemens, Erlangen, Germany) [13]. Pelvic sagittal and axial oblique T_2_-weighted images and para-axial oblique T_1_-weighted images were acquired before and after the administration of intravenous contrast (Dotarem, Guerbet, Villepinte, France: 0.1 mmol gadolinium per kg of body weight, 3 mL/s injection speed) using a 2 min delay. Pelvic axial oblique diffusion weighted imaging (DWI) was also acquired with the generation of apparent diffusion coefficient (ADC) maps.

One expert radiologist (S.Y.H.) manually segmented the tumor by drawing regions of interest (ROIs) on contrast-enhanced T1-weighted (T1c) and T2-weighted images and the ADC maps separately, aiming at including all viable tumor tissue on the single slice (2D) displaying the largest cross-sectional tumor area. The texture analysis and feature extractions have been reported in detail previously [13]. Briefly, the ROIs were drawn using the commercial research software TexRAD (TexRAD, part of Feedback, Cambridge, UK), and a total of 87 MRI tumor texture features were extracted and included when developing the RPI.

### 2.3. Gene Expression Analyses

Gene expression data were available from previous analyses [24,25]. Briefly, RNA was extracted from fresh frozen tissue after selection of the area of high tumor purity, and hybridized to agilent whole human genome microarrays [24,26]. The data were normalized and microarray intensities were determined using J-Express (J-Express, Molmine, Bergen, Norway) [27]. Expression values were available from 40,325 genes per sample. Feature subset selection (FSS) analysis was performed to identify genes that were significantly differentially expressed in lesions with high versus low RPI, with *p* < 0.01 considered significant, and using a cut-off of 1.7 for fold change.

### 2.4. Model Development and Statistical Analysis

The LASSO analysis using regularized Cox regression mode was performed in the R software version 4.0.0 “*glmnet*” package [28]. The 200-times cross validation and the LASSO regression were employed to select the tumor texture features to be included in the model. Four tumor texture features were selected by the LASSO Cox regression model based on the optimal lambda value for inclusion in the RPI score in the study cohort (n = 95): mean positive pixels on T1-weighted contrast-enhanced series (T1c_Mpp), entropy on T2-weighted series (T2_Entropy), kurtosis on T2-wighted series (T2_kurtosis), and kurtosis on the ADC map (ADC_kurtosis).

The values for the tumor texture features weighted by the coefficients from the LASSO regression generated an RPI for each patient, using the formula where *X_gi_* is the value of the texture feature for patient *i* and *β_g_* is the LASSO coefficient for the target *g*.
$$Radiomic\;Prognostic\;Index_{i}=\overset{n}{\underset{g=1}{\sum }}\beta_{g}\times X_{gi}$$

Receiver operating characteristics (ROC) curve of RPI for predicting disease-specific death was plotted and RPI yielded an area under the curve (AUC) of 0.72 (95% confidence interval (CI): 0.58–0.85; *p* = 0.004) and 0.61 (95% CI: 0.45–0.77; *p* = 0.17) for the patients in the study cohort and MRI validation cohort, respectively. The patients were classified into high RPI or low RPI score according to an optimal RPI cut-off determined by the maximum Youden´s index (defined as the sum of sensitivity and specificity).

ROC analysis and the prognostic value of the derived RPI were analyzed using SPSS (Statistical Package of Social Science) version 25 (IBM, Armok, NY, USA). All *p*-values were two-sided and considered statistically significant when *p* < 0.05. Chi-square test was used to determine associations between the RPI and the clinicopathological variables and for comparison of clinicopathological variables between patient cohorts. Spearman correlation was used for assessing the association between continuous data. For univariate analysis of disease-specific survival, the Kaplan–Meier (log rank test) method was used. Disease-specific survival was defined as the date of primary surgery as the entry date and time to death due to endometrial carcinoma as the endpoint.

## 3. Results

### 3.1. High Radiomic Prognostic Index Predicts Poor Outcome and Aggressive Clinicopathological Features in Endometrial Cancer

Patients with a high RPI score had significantly poorer 5-year disease-specific survival compared with patients with a low RPI (63% versus 92%, *p* < 0.001, Figure 2B). This finding was reproduced in the MRI validation cohort (n = 82), demonstrating significantly lower 5-year disease-specific survival in patients with a high RPI score compared with patients with a low RPI score (66% versus 88%, *p* = 0.030, Figure 2C).

High RPI was significantly associated with clinicopathological characteristics of aggressive endometrial cancer such as deep myometrial invasion in both the study cohort (*p* = 0.001) and the MRI validation cohort (*p* = 0.004). Advanced FIGO stage (stage III–IV) and lymph node metastases were also associated with high RPI in the MRI validation cohort (*p* = 0.03 and *p* = 0.02, respectively). No significant associations were observed between RPI and age, histological subtype (endometrioid/non-endometrioid), or grade (Table 2).

### 3.2. High RPI Identifies Potential Prognostic Genes

Feature subset selection analysis identified a list of 46 differentially expressed genes in patients with high RPI versus patients with low RPI (*p* < 0.001, Figure 3A). A total of 22 upregulated and 24 downregulated genes were identified when using fold change of 1.7 as cut-off. *COMP* (cartilage oligomeric matrix protein) was identified as the most upregulated gene, whereas the most downregulated gene was *DMBT1* (deleted in malignant brain tumors 1). Patients with high *COMP* expression had significantly reduced disease-specific survival in both the study cohort (n = 95; *p* = 0.004; Figure 3B) and the mRNA validation cohort (n = 161; *p* = 0.03; Figure 3D). Similarly, patients with low *DMBT1* expression had reduced disease-specific survival in both the study cohort (*p* = 0.010; Figure 3C) and the mRNA validation cohort (*p* < 0.001; Figure 3E).

## 4. Discussion

This study presents a novel prediction model using MRI radiomics, consisting of tumor texture features, that enables non-invasive prediction of high-risk disease in endometrial cancer patients. Importantly, we have also identified novel links between the radiomic tumor profiles and corresponding specific genetic alterations that also predict outcome. To the best of our knowledge, this is the first study describing clinically relevant links between radiomic tumor profile and molecular and genetic tumor profile in endometrial cancer focusing on prognostication.

We developed a radiomic prognostic index (RPI) based on texture features for prognostication in endometrial cancer. The RPI was based on four tumor texture features selected by the model (“T1c_Mpp”, “T2_Entropy”, “T2_kurtosis”, and “ADC_kurtosis”) using LASSO Cox regression in the study cohort (n = 95), and the RPI was validated in a separate MRI cohort (n = 82). The endometrial cancer patients were then categorized into high and low RPI score groups, and poor disease-specific survival was observed in patients with high RPI. We have previously reported MRI texture data on the same endometrial cancer cohort, finding that the texture feature “T1cKurtosis” independently predicted poor outcome and that the texture features “ADC_Entropy” and “T1c_Mpp” independently predicted deep myometrial invasion and high-risk histological subtype, respectively [13]. Interestingly, only one of these texture features “T1c_Mpp” was selected by the LASSO model to be included in the RPI model. This suggests a large overlap in prognostic power for many of the texture variables, and that the employed model for analyzing the texture variables impacts the derived results. The LASSO Cox regression model used for radiomic features selection and the risk score assessment in the present study is, however, in line with methods used by others to develop RPIs in cancers [21,29,30,31].

Several studies have linked MRI-derived radiomic features to prognosis in other cancer types.

Zhang et al. reported a radiomic-based risk score in 286 ovarian epithelial cancer patients that was highly correlated with classification and prognosis [29]. Recently, a radiomic signature based on twelve MRI-derived features was developed for HER2-positive breast cancer that independently predicted disease-free survival [30]. Interestingly, similar to the texture features included in our RPI for endometrial cancer, entropy and kurtosis were among the tumor texture features included in the radiomic signature reported in breast cancer [30], and kurtosis was included in the radiomic signature for glioblastoma [21]. Thus, the tumor texture features most predictive for outcome in our study seem to be shared across cancer types, suggesting a common link to tumor biology related to, e.g., tumor heterogeneity.

We identified associations between the RPI and gene expression profiles in the corresponding tumors. A list of 22 upregulated and 24 downregulated genes was identified in patients with high compared with low RPI. Several of the upregulated genes are linked to extracellular matrix organization, which is associated with development and progression in cancer [32]. The metalloproteinases (*MMPs*) identified in our study were associated with FIGO stage, lymph node metastasis, myometrial invasion, and unfavorable prognosis in endometrial cancer [33]. Although previous clinical trials of MMP inhibitors were unsuccessful, potential inhibitors targeting angiogenesis-promoting MMP are under development [34]. The downregulated genes in the present study include several regulators of the immune system. Cancer-associated mucins (*MUC*) modulate the immune system by binding to various receptors on natural killer cells, macrophages, and dendritic cells, which results in immunosuppression. Anti-MUCs antibodies can target specific hallmarks of cancer and thus have a potential role in immunotherapy [35].

To underline the link between the RPI, translational signature, and patient outcome, we investigated the potential prognostic value of the top upregulated and downregulated genes. *COMP* was the most upregulated gene in our analysis and patients with high *COMP* expression had significantly reduced disease-specific survival in the study cohort. The prognostic significance of this gene was also reproduced in the mRNA validation cohort. Interestingly, COMP protein expression was studied by immunohistochemistry in two cohorts of breast cancer patients (n = 122 and n = 498) [36]. High expression of COMP was associated with poor prognosis and more invasive diseases in epithelial breast cancer [36]. This finding was confirmed by in vitro experiments finding that cells with high level of COMP exhibit a more invasive phenotype [36]. Moreover, in colon cancer, increased *COMP* gene expression is reportedly associated with poor overall survival in a large patient cohort (n = 286), and cell line studies have shown that high COMP expression is associated with increased proliferation [37,38].

*DMBT1* was the most downregulated gene in patients with high RPI. DMBT1 expression was downregulated at both protein and mRNA level in breast cancer [39]. Similarly, in prostate cancer, loss of DMBT1 protein expression and low mRNA expression were correlated with advanced clinical disease such as local invasion and bone metastasis [40]. Loss of DMBT1 expression was an independent poor prognostic marker for cancer associated death in colorectal cancer patients, and predicted lymph node metastases, distant spread, advanced stage, and high histologic grade [41]. Thus, the finding in our study that *COMP* and *DMBT1* expression is associated with survival in endometrial cancer is supported by previous studies reporting the same for other cancers.

Although our patient cohort is from a single center, and is also limited by moderate patient numbers, our results clearly indicate that the developed RPI derived from single-slice tumor texture features might identify biologically relevant molecular tumor features. These findings illustrate the potential of assessing molecular tumor profiles non-invasively based on the tumor’s radiomic profile. Furthermore, if radiomic signatures are also reflected in targetable molecular alterations or linked to the TCGA molecular subclasses in endometrial cancer [6], this would potentially make a radiomic approach more clinically relevant. However, our findings need validation in larger patient cohorts, and future studies should also include mutational status in the analyses.

## 5. Conclusions

The developed RPI, incorporating four MRI-derived tumor texture features, provides a non-invasive method to predict disease-specific survival in endometrial cancer. In addition, the RPI score predicts transcriptional alterations in the corresponding tumor tissue. Radiomic tumor profiling represents a promising approach for better prognostication and identification of the underlying molecular alterations driving cancer progression. By linking radiomic profile to targetable molecular drivers of tumor progression, this may also prepare the ground for more tailored and targeted treatment strategies in endometrial cancer. However, more studies are needed to further elucidate the link between radiomic- and transcriptomic tumor profiles and how this information can be used to improve endometrial cancer patient care.

## Figures and Tables

**Figure 1 jcm-10-00538-f001:**
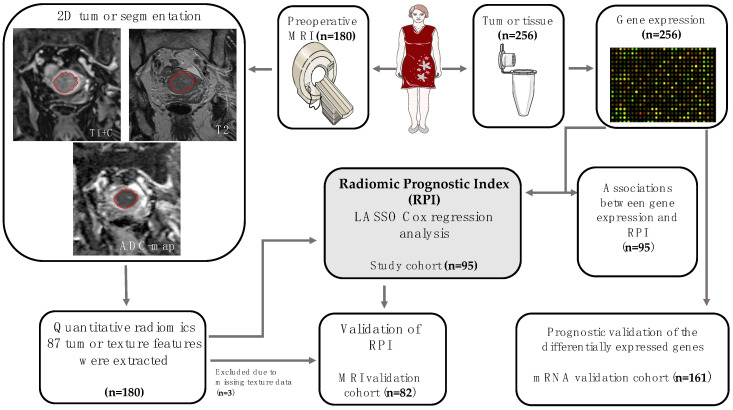
Pipeline for study design and patient cohorts. MRI, magnetic resonance imaging; LASSO, least absolute shrinkage and selection operator; ADC, apparent diffusion coefficient.

**Figure 2 jcm-10-00538-f002:**
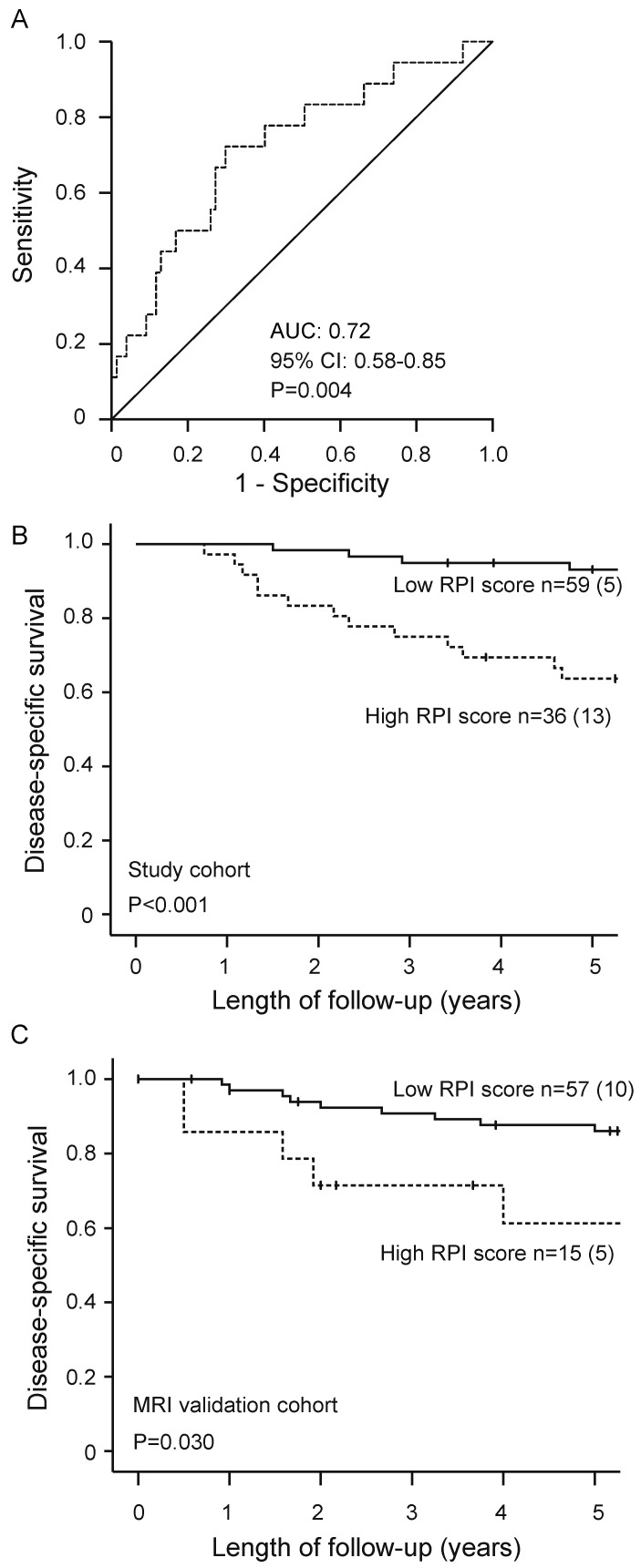
Receiver operating characteristic (ROC) curve for prediction of disease-specific endometrial cancer (EC) death by radiomic prognostic index (RPI) in the study cohort (n = 95; (**A**)). Patients having a high RPI score had significantly reduced survival compared with patients having a low RPI score in both the study cohort (n = 95; (**B**)) and the MRI validation cohort (n = 82; (**C**)). For each category, number of cases/number of cases dying from EC. *p-*values refer to the log-rank test. AUC, area under the curve; CI, confidence interval.

**Figure 3 jcm-10-00538-f003:**
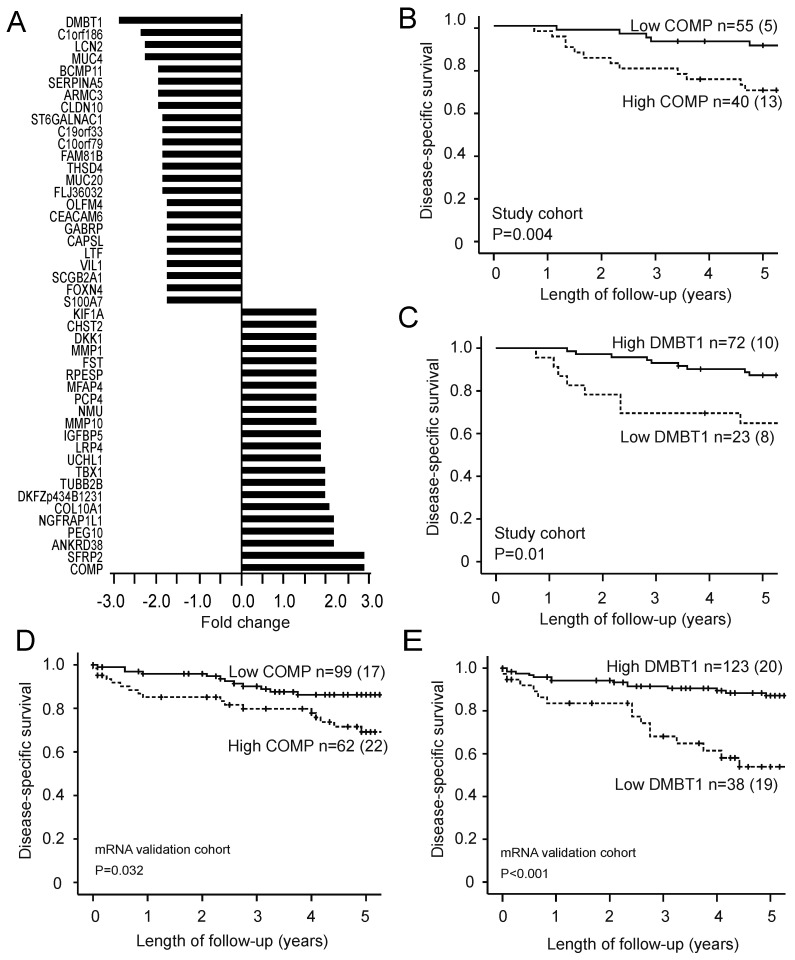
The differentially expressed genes in patients with high RPI (compared with patients with low RPI) were 22 up-regulated and 24 downregulated genes when employing a fold change cut-off of 1.7 (*p* < 0.001 for all; (**A**)). High expression of the most upregulated gene (*COMP*) and low expression of the most downregulated gene (*DMBT1*) are associated with poor disease-specific survival in both the study cohort (*p* = 0.004 and *p =* 0.010 in (**B**,**C**), respectively; n = 95) and the microarray validation cohort (*p* = 0.032 and *p* < 0.001 in (**D**,**E**), respectively; n = 161). For each category, number of cases/number of cases dying from EC. *p-*values refer to the log-rank test. RPI: radiomic prognostic index; COMP: cartilage oligomeric matrix protein; DMBT1: deleted in malignant brain tumors 1.

**Table 1 jcm-10-00538-t001:** Patient and tumor characteristics for the study cohort (having both magnetic resonance imaging (MRI) and mRNA data), MRI validation cohort, and mRNA validation cohort.

Variables	Study Cohort (n = 95)	MRI Validation Cohort (n = 82)	mRNA Validation Cohort (n = 161)	*p-*Value *
**Age, median (range) years**	68 (41–93)	66 (41–84)	66 (38–93)	0.60
**FIGO stage**				0.64
**I–II**	82 (86%)	68 (83%)	122 (76%)	
**III–IV**	13 (14%)	14 (17%)	39 (24%)	
**Histologic subtype**				0.90
**Endometrioid**	76 (80%)	65 (79%)	127 (79%)	
**Non-endometrioid**	19 (20%)	17 (21%)	34 (21%)	
**Histologic grade ^§^**				0.58
**Grade 1**	34 (45%)	30 (46%)	44 (35%)	
**Grade 2**	21 (28%)	24 (37%)	49 (39%)	
**Grade 3**	20 (26%)	10 (15%)	30 (24%)	
**ND**	1 (1%)	1 (2%)	3 (2%)	
**Myometrial invasion**				0.04
**<50%**	47 (50%)	52 (63%)	77 (48%)	
**≥50%**	48 (50%)	28 (34%)	82 (51%)	
**ND**		2 (3%)	2 (1%)	
**Lymph node metastases**				0.69
**No**	76 (81%)	56 (68%)	92 (57%)	
**Yes**	10 (10%)	9 (11%)	23 (14%)	
**ND**	9 (9%)	17 (21%)	46 (29%)	

* Pearson’s chi-square test for categorical variables and Mann–Whitney U test for continuous variables; ^§^ endometroid only; FIGO: The International Federation of Gynecology and Obstetrics; ND, not determined.

**Table 2 jcm-10-00538-t002:** Radiomic prognostic index (RPI) in relation to clinicopathological patient characteristics in the study cohort (n = 95) and the MRI validation cohort (n = 82).

	Study Cohort (n = 95)				MRI Validation Cohort (n = 82)			
	Patients (n)	High RPI (n = 36)	Low RPI (n = 59)	*p-*Value *	Patients (n)	High RPI (n = 15)	Low RPI (n = 67)	*p-*Value *
**Age**				0.54				0.59
**<67 years**	46	16 (44%)	30 (51%)		44	9 (60%)	35 (52%)	
**≥67 years**	49	20 (56%)	29 (49%)		38	6 (40%)	32 (48%)	
**FIGO stage**				0.06				0.03
**I–II**	82	28 (78%)	54 (91%)		67	9 (64%)	58 (88%)	
**III–IV**	13	8 (22%)	5 (9%)		13	5 (36%)	8 (12%)	
**Histologic type**				0.67				0.53
**Endometrioid**	76	28 (78%)	48 (81%)		65	11 (73%)	54 (81%)	
**Non-endometrioid**	19	8 (22%)	11 (19%)		17	4 (27%)	13 (19%)	
**Histologic grade ^§^**				0.16				0.89
**Grade 1–2**	55	17 (47%)	38 (64%)		54	10 (67%)	44 (66%)	
**Grade 3**	39	19 (53%)	20 (34%)		27	5 (33%)	22 (33%)	
**ND**	1	1 (0%)	1 (2%)		1	0 (0%)	1 (1%)	
**Myometrial invasion**				0.001				0.004
**<50%**					52	4 (27%)	48 (72%)	
**≥50%**	47	10 (28%)	37 (63%)		28	10 (67%)	18 (27%)	
**ND**	48	26 (72%)	22 (37%)		2	1 (6%)	1 (1%)	
**Lymph node metastases**				0.31				0.02
**No**	76	27 (75%)	49 (83%)		56	6 (40%)	50 (75%)	
**Yes**	10	6 (17%)	4 (7%)		9	4 (27%)	5 (7%)	
**ND**	9	3 (8%)	6 (10%)		17	5 (33%)	12 (18%)	

* *p-*values from Pearson’s chi-square; ^§^ endometrioid only; FIGO: The International Federation of Gynecology and Obstetrics; ND: not determined.

## Data Availability

The data presented in this study are available on request from the corresponding author (Ingfrid.haldorsen@uib.no)

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
