# Peer review of "An MRI-Based Radiomic Prognostic Index Predicts Poor Outcome and Specific Genetic Alterations in Endometrial Cancer"

_jcm, 2021, doi:10.3390/jcm10030538_

Round 1
Reviewer 1 Report
Dear Author,
I read with great interest the manuscript titled “An MRI based radiomic prognostic index predicts poor outcome and specific genetic alterations in endometrial cancer”.
Authors performed a case-control study on 95 consecutive women with endometrial cancer, proposing a new radiomic score RPI for the prediction of poor disease-specific survival for endometrial cancer, and found a connection among it and cancer-related gene expression.
General considerations:
The study deals with a really interesting topic and the structure of the paper is well thought.
Statistical analysis is correct. I also appreciated the subdivision of results in sections.
The topic represents a new frontier in the preoperative analysis of endometrial cancer, as well as radiomic and transcriptomic are gaining always more importance in oncology generally.
Furthermore, the study is written with a proper language.
I have only some suggestion:
- In methods section you should describe better your analysis of RMI, that leads to RPI development. Furthermore you should note down clear definitions of the four tumor texture features (T1c_Mpp, T2_Entropy, T2_kurtosis and ADC_kurtosis), what do they represent and the way you choose them.
- In methods or results section then you should clarify the cut off between high and low RPI and how did you choose it.
- Why didn’t you analyse the overexpression of SFRP2 that seems of the same interest of COMT from Figure 3A?
- In literature I didn’t find material that deals with COMT an DMBT1 expression in endometrial cancer. On the other hand in your introduction you quoted The Cancer Genome Atlas: PROMISE study and its molecular classification are gaining always more importance in endometrial cancer approach. I thought you should base on the molecular pattern it took under consideration instead of looking for new gene under/over-expressed. I think that a connection among radiomic and that molecular classification should be material for a new study in future, and you should at least propose it or speak about in discussion section.
Author Response
We would like to thank you for your constructive review of our manuscript. We have made a detailed response to all comments and revision of the manuscript is done accordingly. All suggested changes and response to comments are listed below and changes in the manuscript are highlighted using "track changes".
- In the revised manuscript, under Methods section – Model development and Statistical analysis (page 4, lines 141-155), more information about the sequences from which the four tumor features are derived have been included. Furthermore, to increase clarity some of the text describing the model/texture features has been moved from Results to Methods.
The development of the RPI model is also now described in more detail in the same section. Furthermore, we have included data on AUC for RPI for predicting disease-specific death both in the study cohort and in the validation cohort. -
Thank you for pinpointing that this seemed unclear, which may have been due to the fact that this information was not previously given in Methods.
In order to increase clarity, the description on how the e cut–offs were defined for r high and low RPI is now given in Methods (page4-5, lines 150-158).
-
We fully agree with the reviewer that based on Figure 3A SFRP2 overexpression seems to be almost as pronounced as that seen for COMP. In line with this, gene expression for SFRP2 predicted outcome, as did many of the other listed genes in Figure 3A.
However, in order to increase clarity, we decided to include survival plots for only the two genes being most up- and downregulated. This choice hopefully clearly illustrates the concept that the RPI is associated with specific gene-expression profiles and that some of these genes also are prognostic. -
The idea to study radiomic profiles in relation to TCGA molecular classification is highly commendable, however, it would be beyond the scope of this study. However, the clinical relevance of such an approach is highly intriguing, and this is now briefly addressed in the discussion (page 10, lines 314-317).
Reviewer 2 Report
This is a well written manuscript reporting a study aiming to develop and validate a radiomic prognostic index (RPI) derived from textural features extracted from preoperative MRI and to explore possible associations between the RPI and the corresponding transcriptional profile in endometrial cancer patients.
Method is well exposed, results are interesting and tauthors have from the start planned the model and its validation cohort with is mandatory.
As for any model, it would have been better to use a multicentric validation cohort (or from another center), this point is specified in the discussion
Comments
- how patients were assigned to each cohort ? by period of time ? randomly ? by pairing on histological features ?
- What was the AUC for the validation cohort (AUC for prediction of disease-specific endometrial cancer (EC) death by radiomic prognostic index (RPI) in the study cohort was 0.72) ? was it above 0.7 ?
Author Response
We would like to thank you for your constructive review of our manuscript. we have made a detailed response to your comments and revision of the manuscript is done accordingly. All suggested changes and response to comments are listed below and changes in the manuscript are highlighted using "track changes".
-
Three cohorts were used in the study: The study cohort, MRI validation cohort and mRNA validation cohort. Patients were assigned to the cohorts based on available data, meaning that patients with both MRI- and mRNA data constituted the study cohort, patients with only MRI were assigned to the MRI validation cohort and patients with only transcriptional data available were assigned to the mRNA validation cohort. Patient- and tumor characteristics for the study cohort (having both MRI- and mRNA data), MRI validation cohort and mRNA validation cohort are presented in Table 1, comparing the cohorts. The cohorts were designed based on availability of data, and this has now been stated in methods (page 2, line 81-82). The following text explains the information given above. We hope that this is now easier to understand.
-
The AUC for our MRI validation cohort was 0.61 (p=0.17). This information is now given in the methods(page 4, line 150-153). The AUC and also the prognostic value of the RPI appear to be stronger in the study cohort than the MRI validation cohort. This difference is likely due to the well-known “overfitting” of the model in the cohort on which the model is developed.